# Clinical outcomes of chikungunya across age groups: A systematic review

Judith O. M. Bentvelzen[1], Adrianne M. de Roo [2,3]*, Gabriel S. Gurgel do Amaral[1], Sara W. Quist[1,3], Timon Louwsma[1,3], Gerard T. Vondeling[2], Maarten J. Postma[3,4,5,6]

1 Asc Academics B.V., Groningen, the Netherlands, 2 Valneva Austria GmbH, Vienna, Austria, 3 Department of Health Sciences, University Medical Center Groningen, Groningen, the Netherlands, 4 Department of Economics, Econometrics & Finance, Faculty of Economics & Business, University of Groningen, Groningen, the Netherlands, 5 Center of Excellence for Pharmaceutical Care Innovation, Universitas Padjadjaran, Bandung, Indonesia, 6 Division of Pharmacology and Therapy, Faculty of Medicine Universitas Airlangga, Surabaya, Indonesia

* a.m.de.roo@rug.nl

## Abstract

### Background

Chikungunya, caused by a mosquito-borne alphavirus, has emerged as a major public health concern due to its expanding geographical range and debilitating health outcomes in affected populations. Although research on chikungunya is growing, variations in the clinical presentation between age groups remain poorly understood. Understanding these variations is crucial for appropriate diagnoses, treatments, and timely prevention efforts. Therefore, this study aims to assess the clinical presentation of chikungunya across age groups and identify age-related differences and similarities.

### Methodology/Principal findings

We conducted a systematic literature review (17-09-2024) using Embase, MEDLINE, and gray literature, including a risk of bias assessment (Downs and Black, National Institute of Health). Results were descriptively presented. A total of 101 studies reported clinical outcomes in three age groups: children (<18 years), adults (18–64 years), and the elderly (≥65 years), with fourteen directly comparing age groups. The findings highlight substantial differences in disease presentation. Hospitalization and mortality rates were highest among the elderly and younger children, while faster recovery was reported for older children compared to other age groups. Symptom presentation also varied by age, with rash and headache being more common in adolescents and adults than in younger children and the elderly. Fever was more prevalent among adults and the elderly compared with children. Arthralgia was less commonly reported in children and the elderly than in adults, warranting attention for potential misdiagnosis due to atypical disease presentation. Lastly, the occurrence of

**Data availability statement:** All relevant data are within the manuscript and its Supporting information files.

**Funding:** AMR and GTV are Valneva employees, and JOMB, GSGA, SWQ and TL are employees of Asc Academics. Asc Academics has received consultancy fees for this project from Valneva Austria GmbH. The funder had no role in study design and data collection and analysis.

**Competing interests:** I have read the journal's policy, and the authors of this manuscript have the following competing interests: AMR and GTV are Valneva employees and own stock options of Valneva. JOMB, GSGA, SWQ, and TL are employees from Asc Academics. Asc Academics has received consultancy fees for this project from Valneva Austria GmbH. MJP reports grants and honoraria from various pharmaceutical companies, including those developing, producing, and marketing vaccines. MJP holds stocks in Health-Ecore (Zeist, Netherlands) and PAG BV (Groningen, Netherlands), and advises ASC.

chronic chikungunya symptoms appears to increase with age. Yet, the heterogeneity in reporting and scarcity of articles describing clinical features of children and the elderly limit definitive conclusions.

## Conclusions/Significance

Our findings highlight the age-related variations in chikungunya disease presentation, reinforcing the need for tailored diagnostics, clinical management, and prevention strategies in all age groups. By identifying these differences, our study helps guide future research and public health policies aimed at reducing the impact of chikungunya worldwide.

### Author summary

Chikungunya is a viral disease spread by mosquitoes and is known for causing fever, severe joint pain, and sometimes long-term complications. However, knowledge on the variation in disease presentation between age groups remains limited. Here, we systematically reviewed existing research to understand how chikungunya affects children, adults, and the elderly. We found that young children and the elderly are at the highest risk for severe disease, such as hospitalization and death. In contrast, older children tend to recover faster than the other age groups. Differences in symptom presentation were also found, such as fever being more prevalent among adults and the elderly compared with children. Rash and headache were observed more frequently in adolescents and adults compared to young children and the elderly. Also, young children and the elderly were reported to experience joint pain less often, which could increase the chance of misdiagnosis. Furthermore, chronic chikungunya symptoms become more likely with increasing age. Our findings highlight the need for better age-specific healthcare strategies to improve the early detection, treatment, and prevention of chikungunya. By identifying age-related differences in the disease presentation, our study helps guide future research and public health policies aimed at reducing the impact of chikungunya worldwide.

## Introduction

Chikungunya is a viral disease caused by the chikungunya virus (CHIKV), an alphavirus transmitted to humans primarily through the bites of *Aedes (Ae.) aegypti* and *Ae. albopictus* mosquitoes [1]. The disease has drawn global attention due to the rapid geographical expansion of its mosquito vectors, driven by climate change, urbanization, and increased human mobility [2,3]. Emerging research suggests that molecular adaptations in *Ae. albopictus* facilitate effective chikungunya transmission even at lower temperatures [4]. In addition, chikungunya is a well-recognized cause of illness among international travelers, as highlighted by recent reports from the GeoSentinel

network and other travel medicine studies [5,6]. Travelers infected with CHIKV returning to countries where vectors are present may lead to local outbreaks, as seen in Italy [7] and France [8]. As a result, over 75% of the global population now resides in areas at risk for CHIKV transmission, with outbreaks becoming more frequent and widespread [9,10]. These outbreaks have imposed a substantial economic and clinical burden worldwide, totaling $2.8 billion in direct costs, $47.1 billion in indirect costs, and 1,985 disability-adjusted life years (DALYs) globally between 2011 and 2020 [11]. To address this growing threat, the World Health Organization (WHO) has included chikungunya in its Research and Development Blueprint for emerging infectious diseases [12].

Clinically, chikungunya manifests in two distinct phases: acute and chronic [13,14]. The acute phase affects around 75% of infected individuals and is characterized by a sudden onset of high fever, severe joint pain, and rash [15]. The severity of the acute phase can vary widely, with approximately 17% of patients requiring hospitalization due to complications, and in some cases, the disease can lead to fatal outcomes [15]. For approximately 43% of affected individuals, the acute phase is followed by a chronic phase. This phase is characterized by persistent symptoms, including debilitating joint pain, fatigue, and reduced mobility, which may persist for months or even years [10,15,16]. The combined acute and chronic burden of CHIKV imposes substantial economic strain on families and healthcare systems, driven by increased healthcare utilization, productivity losses, and possible long-term disability [10,17,18].

Although variations in the clinical presentation of chikungunya across age groups have been reported, data remain limited, as most studies focus on overall health outcomes without stratifying by age [15]. Still, a few studies have highlighted differences in its impact across age groups, highlighting distinct patterns in symptoms, severity, and associated health risks. For example, clinical manifestations such as arthritis and joint swelling can be significantly less frequent in children than in adults [19]. At the same time, the elderly seem to suffer more from the intensity of acute manifestations [20]. Additionally, literature shows that patients aged 60 or older have a higher mortality risk than younger adults [21], whereas infants and those over 75 years face an increased risk of hospitalization [22]. With two chikungunya vaccines currently with regulatory approval [23–25], understanding age-specific disease burden is even more critical to maximize health impact and prevent severe outcomes. However, a thorough systematic literature review (SLR) of CHIKV manifestation across age groups has not yet been performed, leaving a significant gap in understanding the full scope of the disease's impact and appropriate disease control.

To address this gap, we conducted this SLR to assess the clinical presentation of chikungunya across different age groups. By identifying age-specific patterns and specific vulnerable groups, this SLR aims to synthesize current evidence on age-related differences and similarities in the clinical presentation of chikungunya, to support rapid diagnosis, guide appropriate treatment strategies, and enable timely prevention efforts across all age groups - ultimately helping to reduce the global burden of chikungunya.

## Methods

### Literature search and study selection

This paper builds on a previously conducted SLR that assessed the clinical manifestations of chikungunya [15]. To capture the most recent evidence, we updated this SLR and extracted data on the clinical manifestations of chikungunya across different age groups. The SLR adheres to the Preferred Reporting Items for Systematic Reviews and Meta-Analyses 2020 (PRISMA 2020) guidelines. The updated searches were conducted on MEDLINE In-Process via PubMed.com, and Embase and MEDLINE via Embase.com, restricting the update to records published since the last search (S1 File). Similarly, gray literature searches were performed to capture data from the public domain. To ensure the inclusion of only relevant articles, Population, Intervention, Comparator, Outcomes, and Study Design (PICOS) criteria were applied throughout the search and screening process (S1 File). The SLR was registered with PROSPERO (registration number: CRD42024596800) prior to the initiation of the study.

The outcomes of interest were the acute and chronic illness duration, chronic resolution rates, chronicity rate (reported between six weeks and three months after acute infection), hospitalization frequency (acute and chronic), mortality rate, outpatient hospitalization frequency, symptomatic disease rate, the underreporting factor, as well as the rate and duration of specific symptoms: arthralgia, arthritis, fatigue, fever, headache, joint pain, joint swelling, myalgia, nausea, rash, and vomiting. Despite the conceptual similarity among symptoms, arthralgia, joint pain, and arthritis were reported as separate endpoints because several studies differentiated between them, thus necessitating the inclusion of all terms individually in our analysis. Furthermore, the endpoints 'fever' and 'arthralgia' were frequently used as inclusion criteria in the identified studies, yet were also reported as outcomes, leading to prevalence rates of 100%. To prevent overestimation of the prevalence of these symptoms, such populations were excluded. If prevalence rates were missing, the rate was calculated by dividing the number of participants presenting with that symptom by the total population size. In the results section, we focus on outcomes that were reported at least 25 times across the included study populations. In addition, mortality rate, hospitalization rate, and chronic disease rate were included irrespective of reporting frequency due to their substantial impact on individuals and public health. Outcomes with fewer than 25 reports were summarized in the appendix, including the minimum and maximum prevalence rate or durations reported for each age group.

## Screening and data extraction

All retrieved articles were deduplicated, and titles and abstracts were screened against the PICOS criteria using Rayyan. From the selected articles, full texts were examined for eligibility, followed by detailed data extraction organized by study design, patient characteristics, and outcomes of interest. Due to substantial heterogeneity in study design, age group definitions, and outcome reporting, quantitative synthesis (e.g., pooled estimates or meta-analysis) was not feasible. Therefore, the outcomes of the data extraction were presented in a qualitative, descriptive manner. During data extraction, study inclusion and exclusion criteria, population description and subgroups, CHIKV diagnosis status, comorbidities, population size, age, sex, or co-infection of Zika or dengue were also extracted in addition to the outcomes of interest. The screening process of the update was conducted by three independent reviewers, resolving conflicts through consensus. The risk of bias was determined using the National Institute of Health (NIH) quality assessment tool for observational cohort and cross-sectional studies [26], and categorized as "good" (10–14 points), "fair" (6–9 points), or "poor" (0–5 points) [27]. A modified Downs and Black checklist was utilized for studies with another study design [28]. Each study was classified according to its total score as "excellent (24 to 28 points), "good" (19–23 points), "fair" (14–18 points), or "poor" (fewer than 14 points) [29,30]. Studies labeled with a good or excellent study quality are reported in the results as low risk of bias.

## Definition of age groups

We defined three age categories: children (aged 0–17), adults (aged 18–64), and the elderly (aged ≥65). These categories were chosen based on commonly used clinical and epidemiological categorizations in the literature, which highlight age-related differences in immune response, disease severity, and health outcomes [31,32]. If a study reported sufficient data to categorize the study population into one of the three age groups, the study population was included in the analysis. If a study reported data for more than one study population, each age group was included as a separate population. Populations for which insufficient data were reported for age categorization or for which no outcome of interest was reported were excluded from further analysis in this SLR.

Populations were categorized as children or elderly if 1) the age range of that population fell within the specified age boundaries as defined in the inclusion/exclusion criteria (i.e., inclusion: <18 for youth or ≥65 for elderly), or in population characteristics (i.e., highest reported age: 18 for youth, or lowest reported age: 65 for elderly), or if 2) the median or mean age reported was within the range (e.g., mean/median reported age: 15 for youth, or 70 for elderly). For the

adult category, a stricter classification was applied to reduce the inclusion of studies reporting outcomes for children or elderly. Populations were categorized as adults if 1) the age range of that population fell within the specified age boundaries as defined in the inclusion/exclusion criteria (i.e., inclusion: ≥18 and <65), or in populations characteristics (i.e., lowest reported age: ≥18 and highest reported age <65), or if 2) the median or mean plus and minus the standard deviation (SD) were within the adult age range (e.g., inclusion: >16, mean: 59, SD: 5 was categorized as an adult age group, while inclusion: >16, mean: 59, SD: 6 was excluded because the mean plus SD exceeds the 65 age range). When studies reported outcomes for multiple age groups falling within these criteria, each age group was included as a separate population.

Studies that compared the clinical manifestations of chikungunya across age groups were of special interest, as they provide insights under consistent study conditions. The findings of these studies were presented separately to provide an overview of the current comparative knowledge on age-related differences in disease manifestation.

## Results

### Literature search

The SLR update was conducted on 17 September 2024, which retrieved 1,775 additional records. When combined with the 16,308 records from the previous SLR, this resulted in a total of 18,083 records identified. Of these, 6,839 duplicates were removed before the records were screened by title and abstract (N = 11,244). After exclusion of 9,669 records, 1,575 were screened in phase two. Of these, 1,180 records were excluded, resulting in a total of 395 studies included for age-group categorization. Phase 1 and 2 screening outcomes of the SLR update are available in S1 Table. Study characteristics of the newly included studies were extracted (S2 Table), after which the total of 395 studies were analyzed for age-group categorization. Finally, 101 studies reported outcomes of interest for at least one predefined age group and were therefore included in this study. The literature review process is detailed in Fig 1, which illustrates the PRISMA diagram of all included studies. The PRISMA checklist for systematic reviews is available in S1 PRISMA Checklist.

The majority of the included studies were observational in design (N = 97), and four were experimental. Of the observational studies, cross-sectional (N = 34) and cohort studies (N = 22) were the most frequently reported methodologies.

In the 101 studies reporting outcomes of interest, outcomes of interest were reported for a total of 133 study populations that fell into the predefined age categories. In total, 49 child populations were identified across 34 studies, 73 adult populations across 61 studies, and 11 elderly populations across nine studies. Additionally, 14 of the studies provided data comparing multiple age subgroups within the same study. The quality assessment of newly included studies can be found in S3 and S4 Tables.

### Studies directly comparing age groups

The 14 studies comparing clinical outcomes across age groups indicate that chikungunya manifests distinctly at different ages, among which five studies had a low risk of bias [21,33–36], showing variations in clinical presentation, severity, and rate and speed of recovery (Table 1).

Two studies reported differences in the clinical severity of chikungunya across young children compared to adolescents [21,22]. De Mendonça et al. reported a significant increase in the risk of death from chikungunya among children aged 0–9 years compared to adults aged 20–29 years, while no significant increase in risk was observed when comparing the 10–19 years age group to adults aged 20–29 years [21]. Dorléans et al. reported higher hospitalization rates in infants (younger than 1 year) compared to older children and adults [22]. Differences in disease presentation were also highlighted by studies reporting different symptom rates among children. In infants categorized into age groups of 1–3 months, 3–6 months, 6–12 months, and 12–24 months, hyperesthesia was more frequently observed in infants younger than six months, whereas arthralgia was predominantly reported in children aged 12–24 months, with a higher chance of chronic arthralgia [37]. Warnes et al. found that this trend continues with older children and adults. In their study, they compared

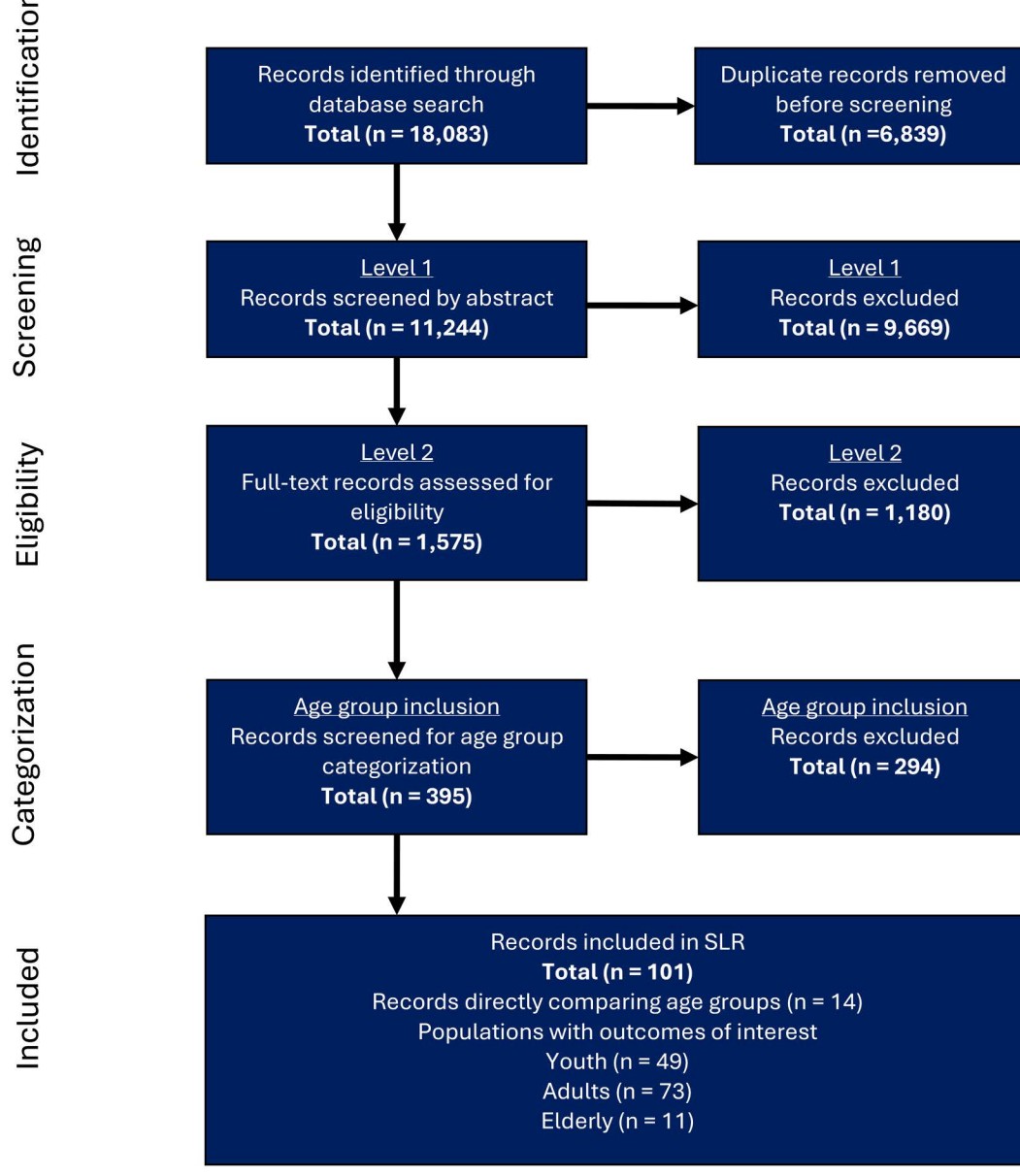

**Fig 1. PRISMA flow diagram of the selection process.** The numbers presented represent the total records included in this synthesis, combining records from the original SLR and the updated search.

children aged 0–4 years with children aged 5–9, children aged 10–15, and adults aged 16–55. They reported that the lowest odds of arthralgia were observed in children aged 0–4 years, and the chance of arthralgia and chronic arthralgia increased with age [36]. Ananth *et al.* described the same pattern for myalgia and shaking chills in a study for children aged 0–4 years, 5–15 years, and older than 15, and both symptoms increased with age [38]. Raghavendhar *et al.* showed that children older than eight presented a higher prevalence of the following symptoms: rash, headache, joint pain, joint swelling, vomiting, rhinitis, hepatomegaly, and restlessness [39]. Further, Ananth *et al.* described how cough was more present in the pediatric population aged 0–4 years compared to children aged five and older [38].

**Table 1. Studies reporting multiple age groups and their key findings.**

| Author | Age groups | Key findings | Study topic | Population | Country |
|---|---|---|---|---|---|
| Manimunda *et al.*, 2010 [33][a] | 10–13<br>14–19<br>20–29<br>30–44<br>≥45 | • Significantly more patients below 20 years had a fever<br>• The cure rate for patients above 45 years was significantly lower | Clinical progression of chikungunya (acute and chronic) | 203 confirmed chikungunya cases aged >10 | India |
| Arif *et al.*, 2020 [40] | Paediatric<br>Adult | • Of paediatric patients, 88.2% were cured (with sequelae: 11.8%)<br>• Of adult patients, 69.9% were cured (with sequelae: 30.4%) | Epidemiology and clinical course of chikungunya | 40 confirmed acute chikungunya cases aged >1 | Indonesia |
| Chopra *et al.*, 2012 [34][a] | <16<br>16-54<br>>54 | • More elderly patients (>54 years) had severe acute illness, and recovery was less quick<br>• Children (<16 years) appeared to suffer less from the intensity of the acute illness. In contrast, elderly patients often looked very sick and remained confined to bed for 4–6 days | Clinical course of acute chikungunya | 509 cases | India |
| Imad *et al.*, 2021 [45] | Children<br>Adults | • Arthritis was significantly more frequent in adults (70%) than in children (10%). Similarly, swelling of the joints and stiffness tended to be more frequent in adults<br>• Ataxia only presented in adults (40%), whereas 30% of children developed seizures | Clinical features of acute chikungunya | 50 confirmed chikungunya cases, children (n = 10) and adults (n = 40) | Maldives |
| Ananth *et al.*, 2020 [38] | 0–4<br>5–15<br>>15 | • Myalgia and shaking chills prevalence increased with age<br>• Cough was more prevalent in the youngest age group (0–4 years)<br>• Vomit was more common in children aged 5–15 | Clinic-epidemiological study of chikungunya symptoms | Confirmed chikungunya cases aged 0–4 (n = 8), 5–15 (n = 52), and >15 (n = 245) | Mexico |
| Economopoulou *et al.*, 2009 [41] | 15–24<br>25–44<br>45–64<br>>65 | • The incidence of atypical cases, severe cases, and mortality increased with age<br>• Incidence of atypical disease was 30 times higher in patients >65 compared to those <45 years<br>• When compared to those aged <40 years, atypical cases aged 40–60 years were 2.5 times more likely to develop severe disease, and those aged >60 years were 1.6 times more likely<br>• Age above 85 years was associated with increased mortality | Atypical chikungunya clinical manifestations and mortality | 610 confirmed chikungunya cases among hospitalized patients | Réunion |
| Gavotto *et al.*, 2019 [37] | 1–3 mo.<br>3–6 mo.<br>6–12 mo.<br>12-24 mo. | • Hyperesthesia was more frequently found in the youngest children (< 6 months), and arthralgia was mostly reported in older children who could walk (> 12 months) | Clinical features of chikungunya | 154 confirmed chikungunya cases in children (aged 1–24 months) | Guadeloupe |
| Godaert *et al.*, 2017 [43] | <65<br>>65 | • A more atypical presentation was found in older individuals.<br>• Severe presentation was higher in older individuals, but not statistically significant<br>• Arthralgia was more present in younger individuals<br>• Hospital stays after emergency admission occurred more often in older individuals | Clinical presentation of acute chikungunya in younger and older adults | 267 confirmed chikungunya cases >65, and 109 cases <65 | Martinique |
| Raghavendhar *et al.*, 2016 [39] | 0–8<br>8–18 | • Older children presented a higher prevalence of the following symptoms: rash, headache, joint pain, joint swelling, vomiting, rhinitis, hepatomegaly, and restlessness | Epidemiological study of chikungunya | 249 confirmed chikungunya cases in children | India |
| Consuegra-Rodríguez *et al.*, 2018 [35][a] | < 40<br>≥40 | • More chronic joint pain and morning stiffness in older adults (>40) | Epidemiology of chronic chikungunya | 65 confirmed chikungunya cases 1 year after acute chikungunya | Colombia |
| Dorléans *et al.*, 2018 [22] | < 1<br>1–4<br>5–14<br>15–44<br>45–59<br>60–74<br>> 75 | • Highest incidence rate of hospitalization in infants (<1) and the elderly (>75)<br>• From 1 year old onwards, the prevalence of severe cases increased with age, and it was more frequently reported in elderly subjects and subjects with underlying health conditions. The mean mortality age was 75 (median 81) | Clinical course of severe chikungunya in hospitalized patients | 1,836 confirmed hospitalized chikungunya cases | Martinique and Guadeloupe |

*(Continued)*

**Table 1.** (Continued)

| Author | Age groups | Key findings | Study topic | Population | Country |
|---|---|---|---|---|---|
| Tun *et al.*, 2022 [44] | 10-17 18–30 31–45 46–60 > 60 | • Maculopapular rash was less prevalent in individuals aged >46 years | Clinical features of chikungunya | 162 confirmed chikungunya cases | Thailand |
| de Mendonça *et al.*, 2023 [21][a] | 0-9 10-19 20-29 20-39 40-49 50-59 > 60 | • Individuals aged 60 years or older were approximately 80 times more likely to die from chikungunya than those in the reference age group (20–29 years)<br>• Compared with the reference age group, the risk of death from chikungunya was also significantly increased in the 0–9 years, 40–49 years, and 50–59 years age groups | Factors associated with mortality from chikungunya | 127 deaths from chikungunya | Brazil |
| Warnes et al., 2024 [36][a] | 0–4 5–9 10–15 16-55 | • Odds of acute arthralgia increased with age, with the lowest odds of arthralgia in the 0–4 group and the highest odds of arthralgia in adults (16+) relative to the 10–15 group<br>• Children exhibited the highest prevalence of post-acute polyarthralgia in their legs, followed by the hands and torso – a pattern not seen among adult participants<br>• As age increased, the proportion of acute cases decreased while the proportion of post-acute or chronic cases increased | Clinical differences between children and adults in post-acute chikungunya-associated arthralgia | 682 confirmed chikungunya cases and 88 clinically probable chikungunya cases | Nicaragua |

[a]Studies with low risk of bias.

While two articles reported higher disease severity among young children compared to older children [21,22], studies comparing children with adults reported lower disease severity for children [22,34,40,41]. Arif *et al.* reported a higher recovery rate for pediatric patients compared to adults [40], Chopra *et al.* reported that individuals younger than 16 years presented with less severe acute illness compared to age groups of 25 and higher [34], and Economopoulou *et al.* reported increasing disease severity when comparing individuals aged 15–24 with older age groups [41]. Similarly, Dorléans *et al.* reported that, with the exception of individuals younger than one year old, the chance of severe chikungunya increased with age when comparing age groups of <1 years, 1–4 years, 5–14 years, 15–44 years, 45–59 years, 60–74 years, and 75 years and older [22]. Differences in symptom rates were also found between children and adults, with two studies reporting less frequent arthritic symptoms in children [19,36]. Conversely, other symptoms were reported more frequently in children [19,33]. Imad *et al.* reported that three out of ten children presented seizures, while none of the 40 adults presented this symptom [42], and Manimunda *et al.* reported a significantly higher fever rate in individuals below 20 years old [33].

For the elderly populations, five studies reported higher disease severity when compared to children and adults [21,22,34,41,43]. Chopra *et al.* reported that the elderly experience more severe acute illness [34], Godaert et al. and Dorléans *et al.* reported a higher incidence rate of hospital admission among older individuals [22,43], and De Mendonça *et al.* and Economopoulou *et al.* reported higher mortality rates in the elderly compared to younger age groups [21,41]. Two studies reported higher atypical disease presentation among that group [41,43], and one study reported a higher frequency of stiffness and chronic joint pain in adults over 40 years compared to those younger than 40 years [35]. In contrast, one study found that arthralgia was less common in individuals over 65 years [43], and another that maculopapular rash was less frequent in adults over 45 [44].

## Studies reporting outcomes for specific age populations

Among the studies reporting disease outcomes for the defined age groups, 20 outcomes of interest were reported at least once for one of the age groups (Fig 2). The most reported outcome was rash (reported in 67 populations), followed

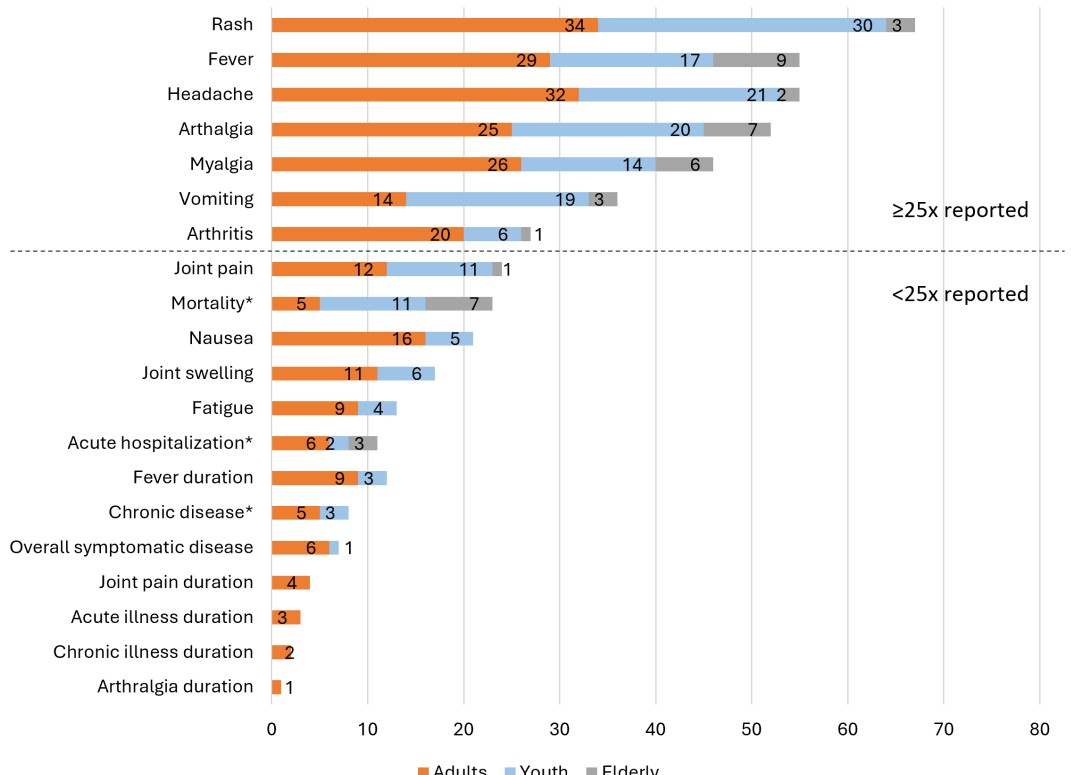

**Fig 2. Number of child, adult, and elderly populations for which the listed outcomes were reported.** The horizontal line divides the outcomes into those reported 25 times or more (which are discussed in the main text) and those reported fewer than 25 times (which are summarized in the appendix). *Mortality, acute hospitalization, and the rate of chronic disease are also discussed in the main text, given their substantial impact on individuals and public health.

by fever and headache (both reported in 55 populations). Six populations for which 100% fever prevalence was reported were excluded because fever was an inclusion criterion in the study. Arthralgia was reported 52 times, with 10 studies reporting 100% prevalence excluded. Other frequently reported outcomes were myalgia, vomiting, and arthritis (reported 46, 36, and 27 times). Mortality, hospitalization for acute chikungunya, and chronicity rate were reported 23, 12, and 8 times, respectively, but will be discussed here given their substantial impact on individuals and public health. The detailed characteristics of the populations associated with these outcomes were reviewed to interpret the data (S5 Table).

In addition to the mortality, acute hospitalization, and chronicity rate, nine outcomes of interest were reported fewer than 25 times (Fig 2). Among these, the duration of joint pain, acute and chronic illness, and arthralgia were only reported for adult populations. The outcomes of interest reported fewer than 25 times were summarized in S6 Table. Further, several outcomes of interest were not reported for any of the predefined age groups, which were the outpatient hospitalization rate, chronic (outpatient) hospitalization, chronic disease resolution, and the duration of arthritis, fatigue, headache, joint swelling, myalgia, nausea, vomiting, and rash.

## Most reported symptoms

Rash was the most reported outcome for children (N=30) and adults (N=34), while it was reported three times for the elderly population (S2 File). The reported rash rates in child populations ranged from 0.0% to 83.3%, with values dispersed across this

range. For 12 out of the 29 studies, a prevalence of 50.0% or higher was found. Among the populations reported in studies with low risk of bias (N = 10), the rash rate was between 4.0% and 75.0%. Among adults, rash prevalence was also widely dispersed, ranging between 3.4% and 93.9%, with 16 out of 34 studies reporting a prevalence above 50.0%. For adults, prevalence in populations from low-risk bias studies (N = 11) ranged from 5.0% to 82.4%. Instead, for the elderly populations, the three reported rash rates were 8.3%, 17.0% and 29.1%, with the first and last derived from studies with low risk of bias [41,46,47].

Fever was reported 29 times for adults and 17 times for children. For elderly individuals, fever was the most reported symptom (N = 9) (S2 File). For nine out of 17 child populations, fever prevalence was below 80.0%, with reported rates ranging from 39.2% to 100%, and from 53.0% to 94.4% when only looking at studies with low risk of bias (N = 4). On the other hand, most studies reporting fever among adults and the elderly reported a prevalence above 80.0%. For twenty out of the 29 included populations, the studies reported a prevalence above 80.0% for adults, with a range between 9.0% [48] and 97.0% [49], or between 9.0% and 96.1% as reported for populations in studies with low risk of bias (N = 9). For elderly individuals, the lowest reported prevalence of fever was 79.4% [50,51], and the other seven studies exceeded the 80.0% with the highest prevalence rate of 96.7% [46]. Within the low risk of bias studies (N = 4), the lowest reported prevalence was 81.8% and the highest was the same.

Headache was reported 21 times for children, 32 times for adults, and two times for the elderly (S2 File). The prevalence among children ranged from 5.2% to 94.1% with a wide dispersion of the reported rates between this range. The majority of the reported headache rates for children were below 60.0% (16 out of 21). Low risk of bias studies reported a rate between 8.7% and 94.1% (N = 8). On the other hand, adult rates mostly reported headache rates of 60.0% or higher (21 out of 32). The range was between 2.9% and 100%, or between 27.5% and 95.8% with low risk of bias (N = 11). The headache prevalence among the elderly was reported in one study with low risk of bias, describing hospitalized cases with atypical chikungunya presentation. It reported a higher rate of 39.0% among 68 patients with non-severe disease, compared to 9.0% in 42 patients who were admitted to the ICU [52].

Arthralgia was reported 20 times for children, 25 times for adults, and seven times for the elderly (S2 File). In all three age groups, the majority of studies identified an arthralgia prevalence exceeding 60.0%, namely 13 out of 20 populations for children, 23 out of 25 for adults, and six out of seven for the elderly. The reported range for children was between 0.0% and 98.6%, and between 60.0% and 92.7% among the studies with good study quality (N = 5). For adults, this rate ranged between 0.0% and 98.5%, which is still the case when solely looking at studies with low risk of bias (N = 6) [34]. Among the elderly, one study reported a rate of 44.1%, three reported a rate between 60.0% and 70.0%, and three low-risk-of-bias studies reported a rate between 80.0% and 83.3%. Most arthralgia rates reported for adults (14 out of 25) exceeded the highest arthralgia prevalence reported for elderly populations (83.3%).

Other frequently reported outcomes were myalgia, vomiting, and arthritis, with high variation in the reported prevalence rates within the age groups. Myalgia was reported 14 times for children, 26 times for adults, and six times for the elderly (S2 File). The prevalence rate for children ranged between 11.6% and 93.0% with eight out of 14 reporting a prevalence of 50.0% or higher. The prevalence for adults ranged between 15.7% and 100%, with 24 out of 26 populations for which a prevalence of 50.0% or higher was reported. On the other hand, the prevalence rate for all elderly populations was below 50.0%. Vomiting rates were most often reported for children (N = 19), as opposed to adults (N = 14) and the elderly (N = 3) (S2 File). The rates among children ranged between 4.3% and 80.8% and were generally higher than the adult populations, ranging between 0.9% and 55.6%, and the elderly populations, ranging between 18.0% and 30.9%. Arthritis was reported six times for children, twenty times for adults, and once for the elderly (S2 File). For children, arthritis rates ranged between 0.0% and 60.0%, with the rates dispersed within this range. Instead, adult arthritis rates ranged between 12.0% and 100%, with seven studies reporting a prevalence rate above 60%. The study reporting on arthritis for the elderly found a prevalence rate of 12.7% [47].

## Mortality

Mortality rates were reported 11 times for children, five times for adults, and seven times for the elderly (S3 File). Prevalence ranged between 0.0% and 17.6% among children. Five studies reported a rate below 1.0%, another five reported

a rate between 1.0% and 10.0%, and one study reported the highest rate of 17.6%, which was reported for 17 patients coinfected with scrub typhus. In the same study, a mortality rate of 4.0% was reported for a group of 25 patients without coinfection [53]. Of the five populations for which the adult mortality rate was reported, three studies reported a mortality rate of 0.0% [54–56], one study reported a rate of 1.9% [57] and one a rate of 3.0% [58]. While most mortality rates reported for child and adult populations were below 1.0%, the lowest reported mortality rate for the elderly was 2.0% and the majority was above 20.0% (4 out of 7). The highest mortality prevalence of 36.7% was observed among 60 patients who were hospitalized in the intensive care unit [46]. The reported mortality in studies with low risk of bias ranged between 0.1% and 17.7% among children (N = 5) and between 2.0% and 36.7% among elderly (N = 5). No such studies reported mortality among adults.

## Hospitalization

Acute hospitalization rates were reported twice for young populations, seven times for adults, and three times for elderly populations (S3 File). Among children, one study reported a hospitalization rate of 8.7% in children below 15 years [59], while another study reported a rate of 30.0% among ten children below 18 years with fever, and all three of the children with fever were also presenting with encephalitis [19]. Neither study was low risk of bias. For adult populations, two studies reported a hospitalization rate of 0.0% [60,61] and the other four studies ranged between 5.0% and 25.0% [55,57,62,63]. The highest hospitalization rate of 25.0% was reported for a group of eight individuals traveling internationally, of which one was hospitalized abroad and one at home. Two studies with low risk of bias reported hospitalization rates, one of 0.0%, and one of 21.1% [60,62]. Among the elderly, one study reported a hospitalization rate of 9.5% [41], while the other two reported hospitalization rates above 50.0% (51.7% and 70.1%) [43,50]. No study among the elderly presented low risk of bias.

## Chronic disease

Chronic disease rates were reported for three child populations, five adult populations, and no elderly populations (S3 File). For the child populations, the three populations were all reported by Warnes *et al.*, a low-risk-of-bias study. They found a chronic disease rate of 16.7% for children aged 0–4 years, 19.7% for those aged 5–9 years, and 23.5% for children aged 10–15 years, 90 days after acute infection [36]. The reported chronic disease rates among adults were higher than those reported for child populations, ranging between 25.3% and 90.0%, or between 35.9% and 90.0% when restricted to low-risk-of-bias studies. The highest chronic disease rate of 90.0% was reported at three months of illness among 112 patients who presented with fever, arthralgia, and rash during the acute phase [64].

## Discussion

This study was conducted to investigate age-related differences in chikungunya presentation. To our knowledge, this is the first SLR to assess these variations across different age groups comprehensively. Our findings confirm that chikungunya can manifest differently in children (<18 years), adults (18–64 years), and the elderly (≥65 years). The results reveal age-related differences in disease severity, symptom presentation, the occurrence of chronic symptoms, and the recovery time. Regarding disease severity, hospitalization and mortality rates were highest among the elderly and young children, reinforcing their status as particularly vulnerable populations. In addition, lower disease severity was reported among older children compared with adults, and fever was more prevalent among adults and the elderly compared with children. In addition, the symptoms rash and headache were observed more frequently in adolescents and adults compared to young children and the elderly. Further, arthralgia was most prevalent in adults, and the chance of developing chronic symptoms was reported to increase with age. Findings reported by studies with high study quality showed consistent findings, although the evidence base from high-quality studies is still underdeveloped.

Our findings align with prior studies indicating that young children and older adults face the highest risk of severe chikungunya outcomes. An SLR of mortality risk factors by Micheleto *et al.* identified increased age (≥60 years) and comorbidities as significant risk factors for severe and fatal chikungunya cases [65]. Other reviews similarly reported that age and pre-existing conditions are key predictors of hospitalization, disease severity, and mortality [66,67]. Likewise, an SLR focused on chikungunya in children identified infants (<1 year) as a high-risk group for severe outcomes [68]. The increased vulnerability in these populations likely stems from the immune system immaturity in infants and age-related immune decline in older adults, with comorbidities further exacerbating disease severity. Recognizing these age-related vulnerabilities is essential for appropriate prevention strategies, early identification, and appropriate clinical management in high-risk groups.

Moreover, the lower reported prevalence of arthralgia in children and the elderly also warrants further attention. As the presentation in these age groups might not include the typical symptoms as established in WHO's regular case definition (which emphasizes fever and joint pain as key diagnostic criteria), this discrepancy may increase the risk of misdiagnosis in these age groups [43,69]. This concern is further supported by studies reporting a higher frequency of atypical disease presentations among the elderly [41,43]. To address this, suggestions have been made to adapt the WHO case definition for the elderly [70]. Such misdiagnosis and misclassification may also partly explain the lower number of studies and underreporting in these age groups within our review, highlighting the need for more age-sensitive diagnostic criteria and treatment approaches.

While infants and the elderly face a higher risk of severe outcomes, including hospitalization and mortality, a higher absolute number of cases is reported for adults [19]. In line with this finding, our review identified more studies reporting clinical outcomes among the adult group. Arthralgia has higher prevalence rates in adults than in children or the elderly, and, in many cases, adults also develop chronic arthralgia that leads to long-term disability and reduced quality of life [71–74]. As the primary working population, this morbidity results in considerable productivity losses and broader economic consequences [11,66]. Understanding age-specific disease burden is essential for designing effective vaccination strategies: vaccinating infants and older adults can reduce the burden on healthcare systems, while targeting working-age adults can mitigate economic losses. An optimal strategy should balance individual health risks with wider public health and economic impacts.

Further differences in symptom presentation across age groups were reported in studies directly comparing age groups. Symptoms such as myalgia, shaking, chills, and headache were more commonly reported among adolescents than in younger children, potentially impacting daily activities, school attendance, and overall well-being [68]. Similarly, rash and headache prevalence rates were higher among adults than among children, aligning with previous reviews [69]. In addition to acute symptom differences, the occurrence of chronic symptoms – particularly persistent arthralgia – was found to increase with age. Borges Costa *et al.* confirmed that older patients are more likely to experience prolonged symptoms, reinforcing the importance of better recognition and management of chronic chikungunya in this population [66]. Supporting this, Lindo *et al.* reported that a reduced immune response during acute infection in the elderly was linked to an increased risk of chronic disease [75].

Our study has some limitations. A high degree of clinical and methodological heterogeneity among the included studies prevented the conduct of a quantitative synthesis and limited the direct comparability of outcomes across studies. While this SLR offers a comprehensive overview of the available evidence, the scarcity of articles describing clinical features of children and the elderly reduces the reliability of current findings for subgroups. Future research could benefit from applying more specific inclusion and exclusion criteria to allow for fair comparison and quantitative synthesis. Additionally, a further subdivision of age groups (e.g., distinguishing between children <12 years and adolescents 12–18 years) could provide a more detailed understanding of age-related differences. Another limitation of this SLR is the frequent reliance on symptomatic disease as an inclusion criterion in many reviewed studies. Symptoms such as fever and arthralgia were often required for study inclusion, potentially leading to underreporting of atypical or asymptomatic cases. This

may be particularly relevant for the elderly, who may present with subtler or atypical symptoms, increasing the risk of misdiagnosis.

Previous literature has identified genetic variations in clinical presentation [76]. To maximize the number of studies available for age group comparisons, this study did not include genotype reporting as a selection criterion. However, the genotype may play a role in the clinical presentation of chikungunya in different age groups. For example, the Asian lineage is associated with more asymptomatic disease than non-Asian epidemics [77], and the East/Central/South African lineage comes with higher rates of chronic disease compared to the Asian lineage [16]. Even within the same country, differences in clinical severity between outbreaks have been observed, which may reflect shifts in circulating genotypes [78]. A link between the lineage and clinical presentation between age groups is so far not established. We therefore recommend that future studies report and analyze genotype data alongside patient characteristics to improve age-specific clinical understanding and inform prevention and treatment strategies.

A key strength of our study is the systematic subpopulation analysis, which enables a comparison of the total body of evidence on age-specific disease presentation of chikungunya. By extracting results for all subpopulations from individual studies identified through a systematic search, we consolidated a complete synthesis of the available evidence across groups while minimizing selection bias and improving comparability.

## Conclusion

This SLR highlights significant age-related differences in chikungunya outcomes, emphasizing that younger children, including infants (<1 year), and the elderly (≥65 years) are at the highest risk for severe disease, hospitalization, and mortality. While these groups experience worse clinical outcomes, they may present with fewer typical symptoms, such as rash and arthralgia, compared to adults. Our findings also reinforce that chikungunya significantly impacts all age groups. Adults, while generally at lower risk of hospitalization and mortality than young children and the elderly, frequently experience debilitating symptoms such as arthralgia and the chance of chronic joint pain increases with age, which can lead to substantial morbidity and long-term quality-of-life impairment.

These insights underscore the need for age-specific clinical management and adapted diagnostic criteria to prevent misdiagnosis. Optimizing prevention efforts, such as ensuring broad vaccine coverage across different age groups while prioritizing high-risk populations, is essential to mitigating the impact of chikungunya. Additionally, standardizing data collection and reporting will be key to refining our understanding of chikungunya's clinical burden across age groups and improving evidence-based public health strategies. Future research should further explore age-specific disease trajectories and long-term outcomes to guide more effective interventions. Implementing targeted yet inclusive prevention strategies will be critical to reducing disease burden and improving patient care worldwide.

## Supporting information

**S1 File. Search string and PICOS criteria.**
(DOCX)

**S2 File. Reported prevalence rates of most reported outcomes.**
(DOCX)

**S3 File. Reported prevalence rates of mortality, hospitalization and chronic disease.**
(DOCX)

**S1 PRISMA Checklist. Checklist for systematic reviews.** Page MJ, McKenzie JE, Bossuyt PM, Boutron I, Hoffmann TC, Mulrow CD, et al. The PRISMA 2020 statement: an updated guideline for reporting systematic reviews. BMJ 2021;372:n71. https://doi.org/10.1136/bmj.n71.
(DOCX)

**S1 Table. Phase 1 and 2 screening outcomes.**
(XLSX)

**S2 Table. Study characteristics for age-group categorization.**
(XLSX)

**S3 Table. Downs and Black assessment of included studies.** This checklist was used for included studies with a study design other than observational cohort and cross-sectional studies.
(XLSX)

**S4 Table. NIH assessment of included studies.** This checklist was used for observational cohort and cross-sectional studies.
(XLSX)

**S5 Table. Detailed population characteristics.** Study population characteristics for frequently reported outcomes (≥25 studies), plus mortality, hospitalization, and chronic disease.
(XLSX)

**S6 Table. Summary of outcomes reported in fewer than 25 studies.**
(DOCX)

## Acknowledgments

We would like to thank the internal teams of Asc Academics who helped during the data extraction phase of the SLR, as well as Roma Kwiatkiewicz from Asc Academics for providing medical writing support.

## Author contributions

**Conceptualization:** Judith O. M. Bentvelzen, Adrianne M. de Roo, Gerard T. Vondeling.

**Data curation:** Judith O. M. Bentvelzen.

**Investigation:** Judith O. M. Bentvelzen, Adrianne M. de Roo, Gabriel S. Gurgel do Amaral, Sara W. Quist, Timon Louwsma.

**Methodology:** Judith O. M. Bentvelzen, Adrianne M. de Roo, Gabriel S. Gurgel do Amaral.

**Project administration:** Judith O. M. Bentvelzen, Adrianne M. de Roo.

**Supervision:** Gerard T. Vondeling, Maarten J. Postma.

**Visualization:** Judith O. M. Bentvelzen.

**Writing – original draft:** Judith O. M. Bentvelzen, Adrianne M. de Roo, Gabriel S. Gurgel do Amaral.

**Writing – review & editing:** Adrianne M. de Roo, Gerard T. Vondeling, Maarten J. Postma.

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
