## [Decision Letter · Decision Letter 0]

3 Jul 2025

PNTD-D-25-00648

Clinical Outcomes of Chikungunya Across Age Groups: A Systematic Review

Dear Dr. de Roo,

Thank you for submitting your manuscript to PLOS Neglected Tropical Diseases. After careful consideration, we feel that it has merit but does not fully meet PLOS Neglected Tropical Diseases's publication criteria as it currently stands. Therefore, we invite you to submit a revised version of the manuscript that addresses the points raised during the review process.

Please submit your revised manuscript within 60 days Sep 01 2025 11:59PM. If you will need more time than this to complete your revisions, please reply to this message or contact the journal office at plosntds@plos.org. Please include the following items when submitting your revised manuscript:

We look forward to receiving your revised manuscript.

Kind regards,

Ronald P van Rij

Guest Editor

Georgios Pappas

Section Editor

Shaden Kamhawi

co-Editor-in-Chief

Paul Brindley

co-Editor-in-Chief

**Additional Editor Comments :**

Thank you for submitting your manuscript to PLOS NTD for review. You manuscript has been assessed by two reviewers, whose comments you will find below. Based on these comments, we invite you to submit a revised version in which you address the reviewers comments and suggestions. Especially Reviewer #1 has valuable suggestions to incorporate data on CHIKV genotype and quantitative syntheses of the data that should be addressed in the revision.

**Reviewers' Comments:**

Reviewer #1: The manuscript, entitled 'Clinical Outcomes of Chikungunya Across Age Groups: A Systematic Review', by Judith Bentvelzen et al., is a timely and well-structured systematic review of age-related differences in the clinical outcomes of chikungunya virus (CHIKV) infections. The categorisation of clinical outcomes in children, adults and the elderly fills a significant gap in the literature and is highly relevant to diagnostics, clinical management and public health policy. However, there are four major and one minor issue that should be addressed to strengthen the manuscript's rigour and impact.

Major concerns:

Firstly, the manuscript does not explore potential differences in clinical presentation linked to the infecting CHIKV genotype (e.g. ECSA, Asian or Indian Ocean lineage). Given the evidence that certain genotypes (e.g. IOL) are associated with an increased risk of chronic arthralgia or atypical presentations, this is a significant omission. At the very least, it would be valuable to include a discussion on this topic, even if data could not be extracted from most studies. While the review is systematic and descriptive, the presentation remains largely narrative. Key outcome comparisons (e.g. the prevalence of arthralgia or fever across age groups) could benefit from basic quantitative synthesis, such as forest plots or weighted averages, even if formal meta-analysis is precluded by heterogeneity. The review could address the diagnostic challenges posed by atypical presentations, especially in elderly patients, more effectively. This could explain underreporting or misclassification in older age groups, which has important implications for case definitions and surveillance strategies. While the manuscript acknowledges high heterogeneity, further efforts could be made to analyse the impact of variable diagnostic criteria, symptom definitions or inclusion thresholds (e.g. studies requiring fever for enrolment). Sensitivity analyses or stratification by study quality could strengthen the interpretation of the findings.

Minor concern:

The methods could be clarified to avoid the use of overlapping terms such as 'arthralgia', 'joint pain' and 'arthritis'. Were these harmonised, or were they reported as distinct endpoints?

Reviewer #2: This is a welcome paper at a time when two chikungunya vaccines have been licensed and policy makers are still trying to get their heads around the optimal use of such vaccines, which includes age specific use. This paper shows what we already know in some way, but now describes it with more clarity and more data: Chikungunya affects age groups differently: young children and the elderly face higher risks of severe disease, while older children recover faster. Symptom patterns vary by age, highlighting the need for age-specific healthcare strategies and further research.

The authors did a systematic review and used the PRISMA approach.

Introduction:

May I suggest that the authors also highlight, within the epidemiological overview, that chikungunya is a common infection among international travellers? To support this, the following references could be considered for inclusion:

Chikungunya infection in returned travellers: results from the geosentinel network, 2005-2020.

Bierbrier R, Javelle E, Norman FF, Chen LH, Bottieau E, Schwartz E, Leder K, Angelo KM, Stoney RJ, Libman M, Hamer DH, Huits R, Connor BA, Simon F, Barkati S; GeoSentinel Network. J Travel Med. 2024 Mar 1;31(2):taae005. doi: 10.1093/jtm/taae005. PMID: 38195993

Chikungunya: risks for travellers.

Simon F, Caumes E, Jelinek T, Lopez-Velez R, Steffen R, Chen LH. J Travel Med. 2023 Apr 5;30(2):taad008. doi: 10.1093/jtm/taad008.

In terms to climate change and increased vectorial capacity, maybe the authors could consider adding this reference in addition to the one by Liu/Rocklov:

Climate change and vector-borne diseases: a multi-omics approach of temperature-induced changes in the mosquito.

Bellone R, et al. Failloux AB. J Travel Med. 2023 Jun 23;30(4):taad062. doi: 10.1093/jtm/taad062.

Discussion:

While the authors' findings are consistent with previous studies showing that young children and older adults are at highest risk for severe chikungunya outcomes, further discussion is warranted. Given that chikungunya infection is significantly more prevalent in adults—leading to substantial morbidity despite relatively low mortality—the overall DALY (Disability-Adjusted Life Years) burden and socio-economic impact may in fact be greater in this age group than in the smaller populations of the very young and very old. Although the latter may have higher relative mortality, the absolute numbers are comparatively lower.

There appears to be a strong focus in the current discourse on prioritizing vaccine use for the very young and the elderly. However, the authors should consider whether this truly represents the most optimal vaccination strategy. A discussion on this point would be valuable, and the authors are encouraged to stimulate further work in the form of economic evaluations, DALY assessments, and modeling studies to guide future policy decisions.

Regarding vaccines, I would also recommend that the authors expand their reference list to include more literature on chikungunya vaccines, including the following:

From bench to clinic: the development of VLA1553/IXCHIQ, a live-attenuated chikungunya vaccine.

Chen LH, Fritzer A, Hochreiter R, Dubischar K, Meyer S. J Travel Med. 2024 Oct 19;31(7):taae123. doi: 10.1093/jtm/taae123. PMID: 39255380

**Journal Requirements:**

1) Your manuscript is a Systematic or Scoping Review. Please change the article type to 'Research Article', noting the change in publication fees associated with Research Articles. 

Note : Research Articles require Data Availability Statement.

2) We note that this manuscript is a systematic review or meta-analysis; our author guidelines require that you upload a PRISMA checklist as supporting information. Information about the PRISMA guidance and blank checklists can be found here: http://www.prisma-statement.org/. Please note that if your study analyzes clinical trials, we cannot proceed with consideration of your article until we have this information.

3) Please provide a complete Data Availability Statement in the submission form, ensuring you include all necessary access information or a reason for why you are unable to make your data freely accessible. If your research concerns only data provided within your submission, please write "All data are in the manuscript and/or supporting information files" as your Data Availability Statement.

4) We ask that a manuscript source file is provided at Revision. Please upload your manuscript file as a .doc, .docx, .rtf or .tex. If you are providing a .tex file, please upload it under the item type u2018LaTeX Source Fileu2019 and leave your .pdf version as the item type u2018Manuscriptu2019.

5) Please upload all main figures as separate Figure files in .tif or .eps format. For more information about how to convert and format your figure files please see our guidelines: 

6) Please provide a detailed Financial Disclosure statement. This is published with the article. It must therefore be completed in full sentences and contain the exact wording you wish to be published.

1) Please clarify all sources of financial support for your study. List the grants, grant numbers, and organizations that funded your study, including funding received from your institution. Please note that suppliers of material support, including research materials, should be recognized in the Acknowledgements section rather than in the Financial Disclosure

2) State the initials, alongside each funding source, of each author to receive each grant. For example: "This work was supported by the National Institutes of Health (####### to AM; ###### to CJ) and the National Science Foundation (###### to AM)."

3) State what role the funders took in the study. If the funders had no role in your study, please state: "The funders had no role in study design, data collection and analysis, decision to publish, or preparation of the manuscript."

4) If any authors received a salary from any of your funders, please state which authors and which funders..

7) Please provide a completed 'Competing Interests' statement, including any COIs declared by your co-authors. If you have no competing interests to declare, please state "The authors have declared that no competing interests exist". Otherwise please declare all competing interests beginning with the statement "I have read the journal's policy and the authors of this manuscript have the following competing interests:"

8) As required by our policy on Data Availability, please ensure your manuscript or supplementary information includes the following: 

This information can be included in the main text, supplementary information, or relevant data repository. Please note that providing these underlying data is a requirement for publication in this journal.

**Figure resubmission:**
---

## [Editor Report · Decision Letter 1]

17 Sep 2025

Dear Ms de Roo,

We are pleased to inform you that your manuscript 'Clinical Outcomes of Chikungunya Across Age Groups: A Systematic Review' has been provisionally accepted for publication in PLOS Neglected Tropical Diseases.

Best regards,

Ronald P van Rij

Guest Editor

Georgios Pappas

Section Editor

Shaden Kamhawi

co-Editor-in-Chief

Paul Brindley

co-Editor-in-Chief

Editorial assessment: Thank you for submitting a revision of your manuscript. The reviewers' comments have been appropriately addressed, and I recommend publication of the manuscript.

---

## [Editor Report · Acceptance letter]

Dear Ms de Roo,

We are delighted to inform you that your manuscript, "Clinical Outcomes of Chikungunya Across Age Groups: A Systematic Review," has been formally accepted for publication in PLOS Neglected Tropical Diseases.

Best regards,

Shaden Kamhawi

co-Editor-in-Chief

Paul Brindley

co-Editor-in-Chief
